# Extracellular Calcium-Induced Calcium Transient Regulating the Proliferation of Osteoblasts through Glycolysis Metabolism Pathways

**DOI:** 10.3390/ijms24054991

**Published:** 2023-03-05

**Authors:** Xiaohang Gao, Xiaohui Di, Jingjing Li, Yiting Kang, Wenjun Xie, Lijun Sun, Jianbao Zhang

**Affiliations:** 1Key Laboratory of Biomedical Information Engineering of Education Ministry, Institute of Health and Rehabilitation Science, School of Life Science and Technology, Xi’an Jiaotong University, Xi’an 711049, China; 2Institute of Sports Biology, Shaanxi Normal University, Xi’an 710119, China

**Keywords:** calcium transient, metabolism, proliferation, osteoblasts

## Abstract

During bone remodeling, high extracellular calcium levels accumulated around the resorbing bone tissue as soon as the activation of osteoclasts. However, if and how calcium is involved in the regulation of bone remodeling remains unclear. In this study, the effect of high extracellular calcium concentrations on osteoblast proliferation and differentiation, intracellular calcium ([Ca^2+^]_i_) levels, metabolomics, and the expression of proteins related to energy metabolism were investigated. Our results showed that high extracellular calcium levels initiated a [Ca^2+^]_i_ transient via the calcium-sensing receptor (CaSR) and promoted the proliferation of MC3T3-E1 cells. Metabolomics analysis showed that the proliferation of MC3T3-E1 cells was dependent on aerobic glycolysis, but not the tricarboxylic acid cycle. Moreover, the proliferation and glycolysis of MC3T3-E1 cells were suppressed following the inhibition of AKT. These results indicate that calcium transient triggered by high extracellular calcium levels activated glycolysis via AKT-related signaling pathways and ultimately promoted the proliferation of osteoblasts.

## 1. Introduction

Bone remodeling occurs throughout our lifetime. This key process begins with bone resorption (initiated by osteoclasts) and is followed by bone formation (initiated by osteoblasts), which together maintain bone structure in response to stress and other biomechanical forces. Bone resorption changes the bone microenvironment, especially by significantly increasing the level of calcium near the resorbing bone [1,2]. This increase in extracellular-calcium levels elevates the intracellular calcium ([Ca^2+^]_i_) concentration in osteoblasts, which regulates many of their functions (including bone formation) by stimulating their proliferation. However, it remains largely unknown how extracellular calcium influences bone formation and the proliferation of osteoblasts.

High levels of extracellular calcium increase [Ca^2+^]_i_ levels in osteoblasts. It has also been confirmed that [Ca^2+^]_i_ is essential for regulating cellular metabolisms functions, such as its effect on glycolytic pathway [3] and tricarboxylic acid cycle (TCA) dehydrogenases (e.g., pyruvate dehydrogenase, α-ketoglutarate dehydrogenase, and isocitrate dehydrogenase) [4,5], and regulation of mitochondrial function [6] by enhancing oxidative phosphorylation [7], F_1_F_o_-ATP synthase [8,9], and electron transport chain (ETC) [10]. Previous studies have shown that energy metabolism is closely related to cell proliferation [11,12,13]. Early reports have indicated that rates of glucose uptake and lactate production are highest during the logarithmic growth of proliferating mouse fibroblasts [14]. Thus, to date, studies have suggested that energy metabolism plays an important role in the regulating of cell proliferation. However, whether [Ca^2+^]_i_-transient osteoblasts proliferation is dependent on changes in energy metabolism has not been studied. A better understanding of how extracellular calcium affects bone formation will inform the development of therapeutics targeting skeletal damage, including osteoporotic bone loss.

In the present study, we aimed to elucidate the effects of high extracellular levels calcium on the proliferation of osteoblasts and determine whether and how these effects are related to energy metabolism.

## 2. Results

### 2.1. Extracellular Calcium Levels Affected Osteoblasts Proliferation but Not Osteoblasts Differentiation

MC3T3-E1 cells were treated with various concentrations (3.6–9 mM) of extracellular calcium for 72 h and their proliferation and differentiation were determined. The 7.2 mM extracellular calcium concentration promoted the proliferation of osteoblasts more effectively than the other concentrations assayed (Appendix A). Thus, this concentration was chosen for all subsequent experiments. Extracellular 7.2 mM calcium significantly increased the proliferation of MC3T3-E1 cells, compared with that of the control group (*p* < 0.05). However, the ALP activity was not significantly different (*p* > 0.05) between the 7.2 mM calcium group and the CON group (Figure 1b).

### 2.2. Metabolomics Analysis

We processed targeted LC-MS data and annotated 36 metabolites, which had appearance reliability in all study groups. The metabolites were present at significantly different levels in the CA and CON groups (Figure 2a). The top 20 metabolites were divided into different clusters (Figure 2b). Our results showed that fructose 1,6-bisphosphate, amino acids (e.g., D-glutamine, L-asparagine, L-serine, and lysine), and products of purine metabolism were potentially significant differentially expressed metabolites. To identify the most significantly differentially expressed metabolites between the CA and CON groups, univariate fold change analysis (FC) was performed with the FC threshold (x) 2 and a *p*-value of <0.05 (false discovery rate adjusted *p*-values). Differences between alterations in the peak intensities of each metabolite in the two groups showed separately using box plots (Figure 2d) for the eight most significant metabolites (*p* < 0.05). Analysis of variance was used to compare the two groups. Pathway analysis of the top altered metabolites showed perturbations in pathways, such as gluconeogenesis, glycolysis, ammonia recycling, pentose phosphate pathway, purine metabolism, and glutamate metabolism. (Figure 2e). We noticed that glycolysis was the metabolic process that was associated with the highest number of differentially expressed metabolites (Figure 2f). Furthermore, to expand the understanding of metabolic pathways related to [Ca^2+^]_i_ change, the module of enrichment analysis was used, which showed that the top 15 additional metabolic pathways including the gluconeogenesis and glycolysis pathway (Figure 2e,f). Together, these results demonstrated that stimulation with 7.2 mM Ca^2+^ increased the rate of glycolysis. In addition, these data indicated [Ca^2+^]_i_ transient induced by 7.2 mM Ca^2+^ altered metabolites levels during osteoblast proliferation.

### 2.3. Effects of Extracellular Calcium on Different Pathways Associated with Energy Metabolism

OCR and ECAR were measured to further identify the specific metabolic between aerobic and anaerobic metabolism. Extracellular 7.2 mM Ca^2+^ promoted glycolysis during osteoblast proliferation. (Figure 3a,c). We next add glucose to release the glycolysis capacity as well as oligomycin to inhibit oxidative phosphorylation, forcing the cells to rely on glycolysis providing energy. We found that ECAR significantly increased in the cells treated with extracellular 7.2 mM calcium. 2-deoxyglucose then reduced the ECAR in all treated cells. We also observed a decreased in oxygen consumption (Figure 3b,d) induced by the addition of extracellular 7.2 mM Ca^2+^. After the injection of oligomycin to inhibit oxidative phosphorylation and the FCCP to uncouple oxidative phosphorylation, we found that the OCR of cells treated with 7.2 mM calcium was largely unaffected. This suggests that cellular metabolism did not rely on oxygen but did rely on an anaerobic pathway. To determine whether proliferation is dependent on changes in energy supply conditions, we examined cell proliferation under different conditions (i.e., glycolysis or only by aerobic oxidation) (Figure 3e). The results showed that the increase in osteoblast proliferation could not be induced when energy was not supplied via glycolysis. We subsequently measured lactic acid production within osteoblasts and in the cell culture medium. The extracellular lactic acid concentrations were significantly higher than the intracellular lactic acid concentrations (Figure 3f). Moreover, the transfer of lactic acid from the intracellular to the extracellular space was significantly higher in the presence than in the absence of high extracellular calcium concentrations. Thus, the cells in the CA group were able to maintain a low lactate environment. Together, these data suggested that an increase in glycolytic energy supply was required to enhance cell proliferation, while an increase in glycolytic energy supply alone will not lead to an increase of cell proliferation.

### 2.4. Effects of Extracellular Calcium on Metabolism Enzymes

To further investigate the involvement of extracellular calcium in the regulation of energy metabolism, expression of proteins implicated in energy metabolism was detected by western blotting analysis. In the glycolytic pathway, PFK is one of the key regulatory of a rate-limiting steps of glycolysis by converting fructose 6-phosphate and ATP into fructose 1,6-diphosphate and ADP [15]. The results showed that expression levels of PFK and LDH, the two key regulatory factors in glycolysis, significantly increased in a time-dependent manner during the extracellular 7.2 mM Ca^2+^ treatment (Figure 4). These data further confirm that the increase in extracellular 7.2 mM Ca^2+^ concentrations promoted osteoblast glycolysis.

### 2.5. AKT Activation Regulated Metabolism via High Extracellular Calcium

MK-2206, which inhibits AKT activity, was used to investigate whether the changes in osteoblast proliferation and metabolism induced by increased extracellular calcium levels were caused by the activation of AKT. As shown in Figure 5, 1 μM MK-2206 prevented the increase in osteoblasts proliferation which was induced by high extracellular calcium levels (Figure 5b). Meanwhile, extracellular calcium promoted the phosphorylation of AKT within 15 min, without affecting total AKT levels (Figure 5a). Moreover, inhibition of AKT significantly down-regulated the expression of PFK (*p* < 0.05) and LDH (*p* < 0.01, Figure 5c,d). These data indicated that AKT activation, which was induced by an increase in extracellular calcium concentration, was required for glycolysis and ultimately for osteoblast proliferation.

### 2.6. Extracellular Calcium Induces Intracellular Ca^2+^ Transients Which Rely on Calcium-Sensing Receptor (CaSR) in Osteoblasts

[Ca^2+^]_i_ concentration was measured using the Fura-2 fluorescence indicator after stimulation MC3T3-E1 cells with extracellular 7.2 mM Ca^2+^. The extracellular 7.2 mM calcium promoted a rapid short spike rise in [Ca^2+^]_i_ concentration (Figure 6a). Moreover, blocking CaSR with 10 μM NPS-2143 abolished the [Ca^2+^]_i_ transients (Figure 6b). These results suggest that the triggering of [Ca^2+^]_i_ transients by extracellular 7.2 mM Ca^2+^ relied on CaSR.

### 2.7. CaSR Promotes Osteoblast Proliferation by Regulating Oxidative Metabolism

CaSR inhibition experiments were performed to determine whether osteoblasts proliferation was affected by CaSR. Although 1 μM NPS-2143 and DMSO did not affect cell proliferation, higher concentration of NPS-2143 inhibited cell proliferation (Figure 7). The proliferation of CA cells in the presence of 10 μM NPS-2143 was significantly lower than in the presence of 7.2 mM Ca^2+^ alone (*p* < 0.01, Figure 7a). These results confirmed that CaSR contributed to cell proliferation, which was induced by extracellular 7.2 mM Ca^2+^.

To further map the function of CaSR in oxidative metabolism, we evaluated the expression of proteins associated with glucose energy metabolism in osteoblasts in the presence of 10 μM NPS-2143. When CaSR was inhibited with 10 μM NPS-2143, the expression of AKT (*p* < 0.05), p-AKT (*p* < 0.01), PFK (*p* < 0.05), and LDH (*p* < 0.01) decreased significantly (Figure 7g–j). This suggests that CaSR inhibition could suppressed AKT phosphorylation, which is required for PFK activation and glycolysis. However, we found that the expression of IDH and OGDH was not significantly affected, suggesting that the inhibition of CaSR didn’t affect the TCA cycle dehydrogenases. These results indicated that CaSR determined the energy source during cellular metabolism, which was connected with the activity of AKT.

## 3. Discussion

Changes in [Ca^2+^]_i_ concentration, triggered by extracellular Ca^2+^ levels, play a key role in osteogenic development and bone remodeling. However, the mechanism underlying the role of [Ca^2+^]_i_ in the regulation of osteoblast proliferation has not been extensively addressed. In this study, a [Ca^2+^]_i_ transient was triggered using 7.2 mM extracellular Ca^2+^. The calcium transient promoted the PFK and LDH expression and increased the rate of extracellular acidification and ultimately osteoblasts proliferation. Moreover, the increased in cell proliferation could be suppressed by the inhibition of glycolysis. Inhibiting AKT with MK-2206 also suppressed the cell proliferation and the expression of PFK and LDH. These data suggested that the increased in osteoblasts proliferation (induced by an increase in extracellular calcium) was regulated by glycolysis and specifically AKT.

Osteoblasts play an important role in the process of bone remodeling throughout our entire lifetime. During the bone remodeling process, the concentration of extracellular Ca^2+^ in the bone microenvironment reaches a high level. Ca^2+^ is a highly versatile second messenger, which controls various cellular functions, including proliferation, differentiation, apoptosis, autophagy, migration, and metabolism [16,17]. In this study, a [Ca^2+^]_i_ transient was triggered by high extracellular Ca^2+^ concentrations (via CaSR) but not the gradual increase in [Ca^2+^]_I_ levels. First, the single-peak transient of calcium may be a signaling mechanism which causes a cascade of intracellular signals. Moreover, this cascade reaction does not occur after the single-peak calcium transient disappears. Studies have reported the existence of numerous “decoder” molecules associated with changes in intracellular calcium transients, such as NF-κB, MAPK, and NFAT [18,19]. Second, when extracellular calcium concentration causes intracellular calcium levels to rise and generate a single-peak transient through CaSR, the transient response of glycolytic enzyme increases inside the cell. This promotes intracellular glycolysis to yield more energy for cell proliferation. The gradual increase in [Ca^2+^]_i_ concentration promotes both glycolysis and aerobic oxidation, but not inclined to the increase of glycolysis. It has been reported that the increase in extracellular calcium levels stimulates DNA synthesis in osteoblasts and boosts their proliferation in vitro [20], which is consistent with the results of this study.

Recent studies have pointed out the factors trigger and maintaining the Ca^2+^ transients of osteoblasts, and how Ca^2+^ signals regulate the proliferation and differentiation of osteoblasts through Ca^2+^ channels are still unknown [21,22]. Although we know that high levels of extracellular calcium trigger intracellular calcium transient, the role of the high calcium microenvironment in osteoblast function remained unclear. Many in vitro studies further showed that early differentiated and mature cells, belonging to the lineage of osteoblasts and chondrocytes, are sensitive to extracellular calcium and express CaSR [23]. It was also established that extracellular calcium triggered an [Ca^2+^]_i_ transient through CaSR. In fact, the activation of CaSR stimulates one of its major effector protein phospholipase C, which hydrolyses PIP_2_ to the diacylglycerol and inositol-1,4,5-trisphosphate (IP_3_) [24]. IP_3_ binds to its receptors at the endoplasmic reticulum (ER) and promotes intracellular Ca^2+^ release into the cytoplasm [25]. The activation of a class I PI_3_K enzyme [26], induce PIP_3_ to bind AKT in response to extracellular signals. The competitive decrease in PIP_2_ levels means that there are not enough IP_3_ molecules to bind to the IP_3_ receptors at the ER, which triggers a short spike in the [Ca^2+^]_i_ transient. Some experimental evidence has suggested that cell proliferation is regulated by high levels of extracellular Ca^2+^ and by subsequent Ca^2+^ entry into the cytoplasm [27,28]. Our results also confirmed that high levels of extracellular Ca^2+^ induced an [Ca^2+^]_i_ transient through CaSR and promoted osteoblast proliferation.

Cell metabolism also affects cell proliferation. The increase in glycolysis process promotes the proliferation of fibroblasts, neuronal cells, mesenchymal stem cells, and osteosarcoma cells, and the up-regulation of genes encoding key enzymes of glycolysis [14,29]. Early reports indicated that rates of glucose uptake and lactate production are highest during the logarithmic growth in proliferating mouse fibroblasts [14]. In this study, the metabolite profiles of MC3T3-E1 cell treated with 7.2 mM Ca^2+^ revealed significant alterations in the metabolite levels which were associated with gluconeogenesis, glycolysis, the pentose phosphate pathway, and purine metabolism pathways. Specifically, the increased in fructose 1,6-bisphosphate and PPRP levels, and the decreased in D-glucose 6-phosphate, L-serine, D-glutamine, lysine, and AMP levels were observed. Moreover, we not only observed impaired mitochondrial biogenesis and decreased mitochondrial oxygen consumption during MC3T3-E1 cell proliferation induced by 7.2 mM extracellular calcium, but also found that these cells exhibited increased ECAR during proliferation, which is in accordance with the previous study [30]. The increase in fructose 1,6-bisphosphate levels indicated that extracellular Ca^2+^ enhanced aerobic glycolysis. The levels of key regulating factor of glycolysis, such as PFK, also increased during the treatment of MC3T3-E1 cells with 7.2 mM. The findings with depleted amino acids in extracellular Ca^2+^ may exhibit excessive consumption of the amino acids in osteoblasts to sustain cell proliferation. Together, these results indicated that extracellular Ca^2+^ enhanced aerobic glycolysis and osteoblast proliferation.

It has been previously demonstrated that AKT is involved in the regulation of glucose uptake and the stimulation of aerobic glycolytic metabolism [31]. Moreover, it has been reported that AKT activation by phosphorylation at Ser-473 is induced by PIP_3_ at the cell membrane [32]. In A-549 cells, the overexpression of Orai1 which attenuates SOCE influx, inhibited cell proliferation [33]. Coincidentally, the inhibition of SOCE by the gene silencing of Orai decreased the levels of phosphorylated AKT, which is consistent with the results of the present study [34]. In the present study, we showed that the [Ca^2+^]_i_ transient promoted osteoblasts proliferation via the phosphorylation of AKT at Ser-473. AKT regulates glucose uptake by increased the transcription and membrane translocation of glucose transporter 1 [35]. Moreover, it was demonstrated that the activation of AKT by phosphorylation positively stimulates aerobic glycolysis [36] and decreases the activity of the PDH-E1α [37]. In this study, we observed that the extracellular-Ca^2+^-mediated activation of AKT not only increased glycolysis but also impaired mitochondrial biogenesis, which was accompanied by decreased mitochondrial oxygen consumption, during the cell proliferation. Consistent with this, another study indicated that the increase in glycolysis was accompanied by a decrease in mitochondrial respiration [38]. Of note, the inhibition of ATK inhibited osteoblast proliferation, but did not affect the calcium transient. This means that after the occurrence of a calcium transient, the activation of AKT was required to promote cell proliferation. We observed that treating MC3T3-E1 cells with 7.2 mM extracellular calcium-induced AKT phosphorylation, promoted cell proliferation and increased the expression of PFK and LDH. A study reported that the activation of AKT increased the activity of PFK and the synthesis of fructose 1,6-bisphosphate [39]. When AKT phosphorylation is inhibited, cell proliferation and glycolytic enzyme expression will therefore decrease. These results indicate that an increase in [Ca^2+^]_i_ transient triggered by high extracellular calcium levels increased the rate of aerobic glycolysis via the activation of AKT and ultimately promoted MC3T3-E1 cell proliferation.

Our study had certain limitations. First, our results showed that the increase in extracellular calcium levels affected cell metabolic processes, such as glycolysis. Although we performed OCR, ECAR, metabolomics, and protein expression analyses, we did not evaluate changes in amino acid and fat metabolism. Second, our results were generated from experiments on MC3T3-E1 cell lines in vitro. Although MC3T3-E1 is a widely used osteoblast cell line [40], which can mimic the physiological activity of typical osteoblasts [41], this study did not conduct animal experiments to verify the effects of high extracellular calcium levels on osteoblasts and bone regeneration in vivo. Third, we were unable to fully elucidate the mechanism underlying the effect of high extracellular calcium levels on osteoblast function, although we demonstrated that intracellular calcium transients can affect osteoblast metabolism and proliferation. Further research, based on ‘multi-omics’ and molecular structural approaches should be undertaken to elucidate the exact mechanism. Despite these limitations, the conclusions of the present study provide new and valuable evidence that the calcium transient triggered by high extracellular calcium levels increased the rate of glycolysis via the activation of AKT to promote osteoblast proliferation.

In conclusion, our current work extended the findings of previous studies and verified that CaSR was involved in the induction of [Ca^2+^]_i_ transients by high extracellular calcium levels, which in turn promoted glycolysis metabolisms via AKT activation, and ultimately increased osteoblast proliferation.

## 4. Materials and Methods

### 4.1. Cell Culture, G0 Cycle Treatment, and Inhibitory Experiment

The osteoblast-like cell line MC3T3-E1 was purchased from the National Laboratory Cell Resource Sharing Platform (Beijing, China). Cells were cultured at 37 °C in a 5% CO_2_ atmosphere in α-minimal essential medium (α-MEM; Gibco; Thermo Fisher Scientific, Inc., Waltham, MA, USA) containing penicillin (50 U/mL), 10% fetal bovine serum (FBS, Gibco; Thermo Fisher Scientific, Inc.), and streptomycin (50 µg/mL). The conditioned medium was changed every 24 h. MC3T3-E1 cell cycle arrest in G0/G1 was achieved by serum deprivation. Cells were washed three times in phosphate-buffered saline (PBS) and cultured in α-MEM containing 2% FBS for 24 h. The cells were then cultured for 72 h to allow for the detection of proliferation, differentiation, and protein levels. Inhibition experiments were performed under inhibitory concentration after serum deprivation. Briefly, for cell proliferation, differentiation, intracellular calcium, and Western blotting, the cells were incubated with the corresponding inhibitors at the specified concentration dissolved in α-MEM. The NPS-2143 (Monmouth Junction, NJ, USA), and MK-2206 was purchased from (Beyotime, Shanghai, China) inhibitors were dissolved in DMSO.

### 4.2. Cell Proliferation Assay, Alkaline Phosphatase (ALP) Activity Assay, and Lactate Concentration Measurements

To evaluate cell proliferation, a Cell Counting Kit-8 (CCK-8, Beyotime) assay was performed according to the manufacturer’s instructions. Briefly, the cells were counted using a cell counter, and 100 µL of the cell suspension (containing 1 × 10^5^ cells) was added to 96-well plates. After the cells adhered to the flat bottom plates, the original culture medium was removed, and 100 µL of culture medium containing the corresponding concentrations of drugs were added according to the experimental grouping. Cells were incubated with α-MEM plus 10% (*v/v*) CCK8 at 37 °C for 40 min and the optical density (OD) of each well was measured at the wavelength of 450 nm using a microplate reader (Molecular Devices Inc., San Jose, CA, USA). The ALP activity test was carried out using the Alkaline Phosphatase Assay Kit (Jiancheng, Nanjing, China) by following the manufacturer’s instructions. Briefly, after 100 µL of a single-cell suspensions (containing 1 × 10^5^ cells) were added to 96-well plates and 100 µL of culture medium containing the corresponding concentrations of drugs was added according to the experimental grouping. Cells were washed with serum-free α-MEM and lysed with 30 μL 0.5% Triton in PBS. Next, 35 μL of the lysed sample was mixed with solutions A and B (50 μL of each) in a 96-well plate and incubated at 37 °C for 30 min. After the addition of solution C, OD values at 520 nm were measured on a multi-plate reader (Molecular Devices Inc.). The concentration of lactate was detecting using the Lactic Acid Content Assay Kit (Solarbio^®^ BC2230), and the emission spectra were measured under a wavelength of 570 nm.

### 4.3. Calcium Imaging

The real-time [Ca^2+^]_i_ concentration was measured using the ratiometric fluorescence Ca^2+^ indicator Fura-2 AM (Beyotime), as previous report [42]. The cells were washed with Hank’s Balanced Salt Solution (HBSS) twice and loaded with 4 μM Fura-2 AM in α-MEM for 30 min in the dark. Fura-2 fluorescence (proportional to [Ca^2+^]_i_ concentration) was visualized under a microscope (IX71, Olympus) at 340 and 380 nm, emitted from a monochromator (Polychrome V, TILL Photonics GmbH, Grafelfing, Germany). The 510 nm emitted fluorescence was detected using a high-speed cooled CCD camera (C9100, Hamamatsu, Japan) and recorded by Simple-PCI software version 6.60. The [Ca^2+^]_i_ concentration was calculated as previously described [43].

### 4.4. Metabolomics Analysis

Metabolomics analysis was conducted by Mtware Biotechnology Co., Ltd. Liquid chromatography-mass spectrometry (LC-MS) data were acquired on a Shimpack UFLC SHIMADZU CBM30A and QTRAP (ABSCIEX, Framingham, MA, USA) system. Samples were resuspended in CH_3_OH at a concentration of 0.1 mg/mL. Injections of 3 μL were performed on a SeQuant ZIC-pHILIC column (5 μm, 2.1 × 100 mm, Waters) with a flow rate of 0.4 mL/min using the following binary solvent gradient of H_2_O (0.1% formic acid added) and CH_3_CN (0.1% formic acid) at 40 °C: initial isocratic composition of 95:5 (H_2_O:CH_3_CN) for 1 min, increasing linearly to 50:50 over 9.5 min, followed by an isocratic hold at 5:95 for 11.1 min, gradient returned to starting conditions of 95:5 and held isocratic again for 14 min. The following settings were used: Heated electrospray ionization (HESI) source temperature, 450 °C; positive ionization mode, 5500 V; negative ionization mode, –4500 V; GS I, 40 psi; GS II, 55 psi; and curtain gas, 35 psi. Multivariate data analysis was conducted on the normalized data in MetaboAnalyst 5.0 (http://www.metaboanalyst.ca, accessed on 21 July 2021). To investigate the relationship between calcium concentration, bone regeneration, and cellular metabolism, we identified the differential metabolic signatures of the two groups of osteoblasts: the control groups (CON) and the application of 7.2 mM calcium group (CA). The metabolic pathway data were analyzed using the MetaboAnalyst and HMDB (http://www.hmdb.ca, accessed on 21 July 2021) databases.

### 4.5. Western Blotting Analysis

The MC3T3-E1 cells were harvested and lysed with RIPA buffers (Beyotime) containing protease and phosphatase inhibitors (Keygentec, Nanjing, China). The total protein concentration was quantified using a BCA kit (Beyotime), and an equal amount of protein was loaded onto each lane of a 10% sodium dodecyl sulfate-polyacrylamide gel and separated by electrophoresis. The proteins were transferred onto a 0.22-µm polyvinylidene difluoride (PVDF) membrane (EMD Millipore, Billerica, MA, USA). Membranes were blocked for 2 h with 5% evaporated skim milk in Tris-buffered saline containing 0.1% Tween 20 and incubated at 4 °C overnight with the primary antibodies. β-actin was used as an internal control. After being washed with TBST, the PVDF membranes were incubated with secondary antibodies (ZSGB-bio, Beijing, China) at room temperature for 30 min, followed by washing with TBST. Finally, the PVDF membranes were then incubated with horseradish peroxidase (HRP)-conjugated secondary antibody IgG (1:8000, Santa Cruz Biotechnology) for 1 h at room temperature. The immune complexes were detected by enhanced chemiluminescence (ECL, CST, Danvers, MA, USA), which revealed the specific protein bands. Primary antibodies against the following targets were used: phosphofructokinase (PFK, ab154804, Abcam, Cambridge, UK), lactate dehydrogenase (LDH, 19987-1-AP, Abcam), pyruvate dehydrogenase (PDH E1 Alpha, 18068-1-AP, Proteintech, Hubei, China), isocitrate dehydrogenase (IDH,123321-1-AP, Proteintech), α-ketoglutarate dehydrogenase (OGDH, ab137773, Abcam), AKT (55230-1-AP, Proteintech), and p-AKT (664444-1-AP, Proteintech).

### 4.6. Oxygen Consumption Rate (OCR) and Extracellular Acidification Rate (ECAR)

The OCR and ECAR of the MC3T3-E1 cells were measured using a Seahorse XFe24 analyzer (Agilent Technologies) by following the manufacturer’s instructions. Briefly, the cells were seeded on a XFe24 Analyzer cell culture microplate at a density of 1 × 10^5^ cells/well. The cells were then cultured for 72 h in medium containing 7.2 mM Ca^2+^; the conditioned medium was changed every 24 h. For the MC3T3-E1 cell mitochondrial stress test, the following concentrations of the inhibitors were used: 1 μM oligomycin, 1 μM FCCP, and 4 μM each of antimycin A and rotenone. For the modified glycolytic stress test, 25 mM glucose, followed by 1 μM rotenone, and finally 50 mM 2-deoxyglucose was used.

### 4.7. Statistical Analysis

Data were expressed as the mean ± standard deviation (SD) and analyzed using Statistical Product and Service Solutions version 22.0 (SPSS; International Business Machines Corp., Armonk, NY, USA). The one-way analysis of variance (ANOVA) and post-hoc Turkey test were used to compare the effects of different cell treatments. Significant differences between mean values were identified in multiple-range tests with a confidence level of *p* < 0.05.

## Figures and Tables

**Figure 1 ijms-24-04991-f001:**
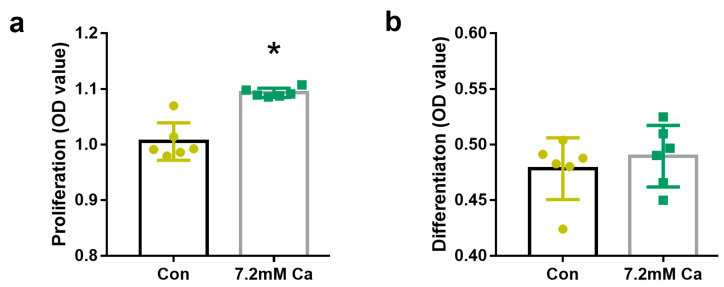
The proliferation and differentiation induced by extracellular 7.2 mM Ca^2+^ in MC3T3-E1 cells. (**a**) The proliferation of osteoblasts. * *p* < 0.05, vs. Con group. (**b**) The differentiation of osteoblasts.

**Figure 2 ijms-24-04991-f002:**
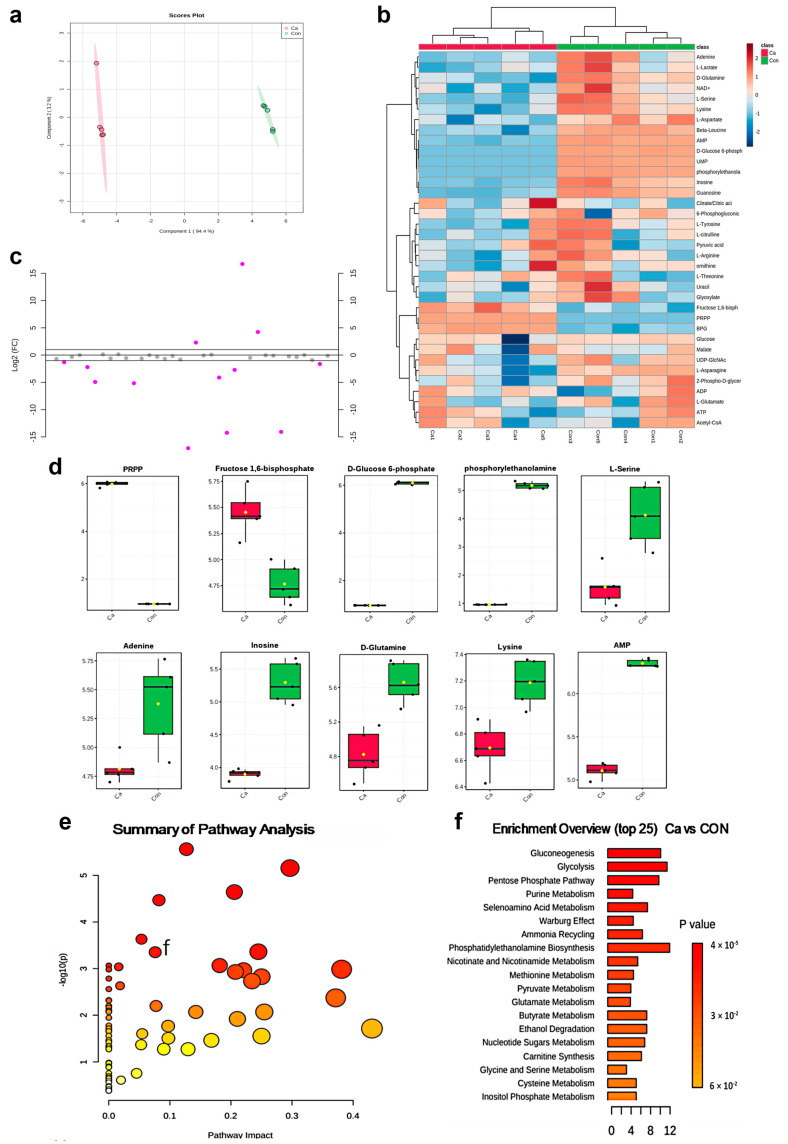
Metabolomics analysis of cells induced by extracellular 7.2 mM Ca^2+^. (**a**) The sparse partial least squares discriminant analysis score scatter plots based on metabolic profiles of class separation based on the top three components of Con and Ca group. (**b**) Heatmap visualization of the 36 discriminative metabolites with hierarchical clustering analysis (HCA) among the Con and Ca group. (**c**) Fold change (FC) score plot of Ca vs. Con. Purples dot represents the discriminative metabolites between two groups. (**d**) Boxplots of the partial eight most significant metabolites (*p* < 0.05) in the analysis of variance results comparing the three groups (Con and Ca group). (**e**) Metabolic pathways are involved in simulation of extracellular 7.2 mM Ca^2+^. Scatter plot presenting enriched metabolic pathways. The color gradient indicates the significance of the pathway ranked by *p*-value (*y*-axis; yellow: higher *p*-values and red: lower *p*-values), and circle size indicates the pathway impact score (*x*-axis; the larger circle the higher impact score). (**f**) Enrichment pathway analysis of extracellular 7.2 mM Ca^2+^ vs. Con group.

**Figure 3 ijms-24-04991-f003:**
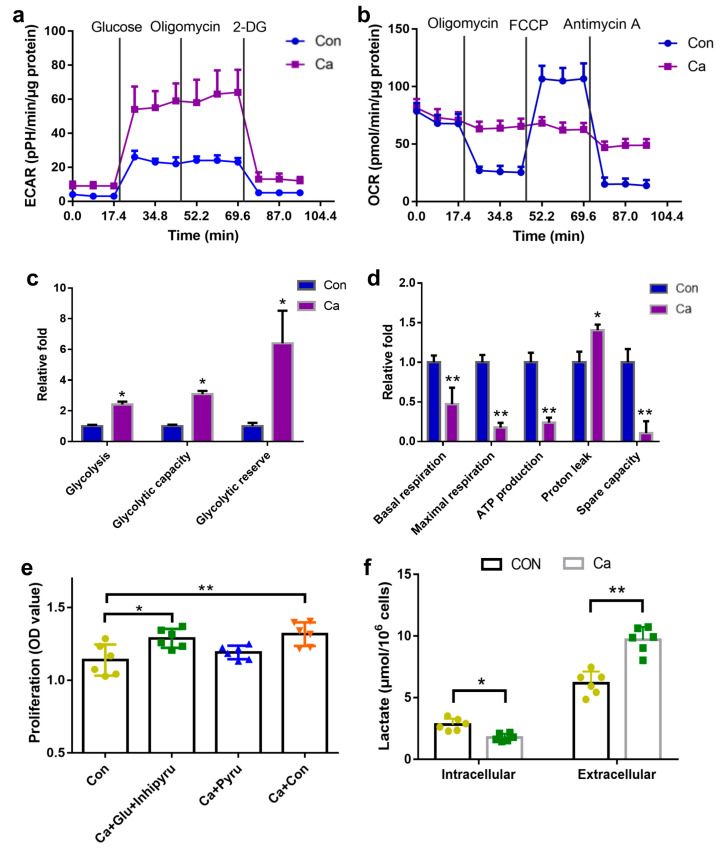
Effect of extracellular 7.2 mM Ca^2+^ on aerobic glycolysis and TCA cycle in MC3T3-E1 cells. (**a**,**c**) ECAR analysis normalized to protein concentration after 72 h treatment in 7.2 mM Ca^2+^. Cells were assayed in XFe24, with sequential injection of 25 mM glucose followed by 1 μM rotenone, and finally 50 mM 2-deoxyglucose. * *p* < 0.05, vs. Ca group. (**b**,**d**) OCR analysis normalized to protein concentration after 3 days cultivation in 7.2 mM Ca^2+^. Following basal respiration, cells were injected sequentially with 1 μM Oligomycin, 1 μM FCCP, and 4 μM Antimycin A. * *p* < 0.05, vs. Ca group. (**e**) The proliferation and differentiation induced by extracellular 7.2 mM Ca^2+^ under the different metabolism pathways for 72 h. * *p* < 0.05, ** *p* < 0.1, vs. Con group. (**f**) Lactate production in MC3T3-E1 cells for 72 h and out of MC3T3-E1 cells for 24 h. * *p* < 0.05, ** *p* < 0.1, vs. Con group.

**Figure 4 ijms-24-04991-f004:**
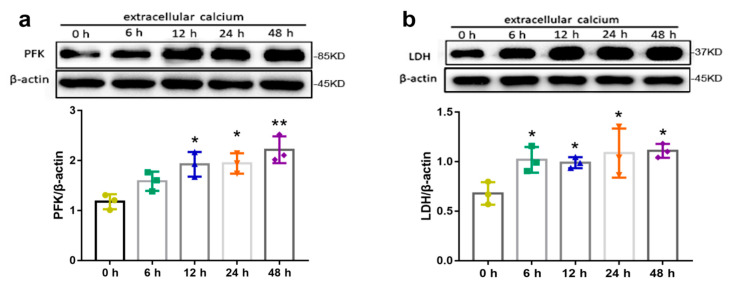
Effects of extracellular 7.2 mM Ca^2+^ on levels of metabolism enzymes. (**a**) PFK and (**b**) LDH. * *p* < 0.05, ** *p* < 0.01, vs. 0 h.

**Figure 5 ijms-24-04991-f005:**
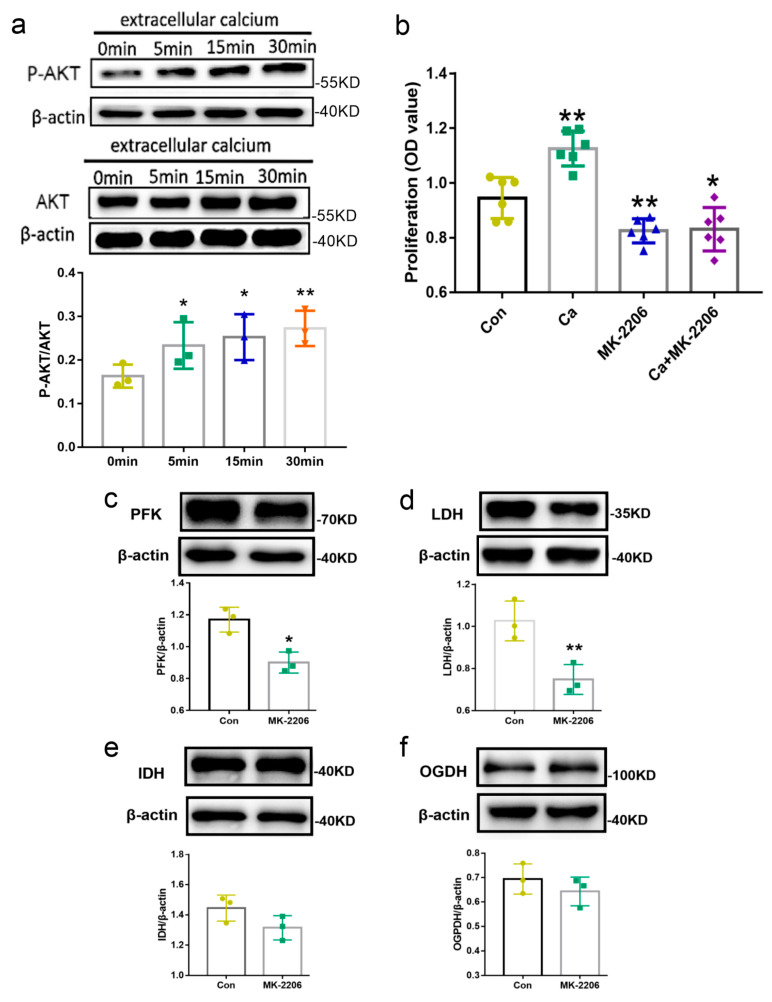
AKT was involved in regulation of energy metabolism. (**a**) Extracellular 7.2 mM Ca^2+^ induced changes of p-AKT and AKT. (**b**) The osteoblasts proliferation treated with 7.2 mM calcium, 1 μM MK-2206, and both of them, (**c**–**f**) Effects of 1 μM MK-2206 and extracellular 7.2 mM Ca^2+^ on PFK and LDH, IDH and OGDH. * *p* < 0.05, ** *p* < 0.01 vs. Con.

**Figure 6 ijms-24-04991-f006:**
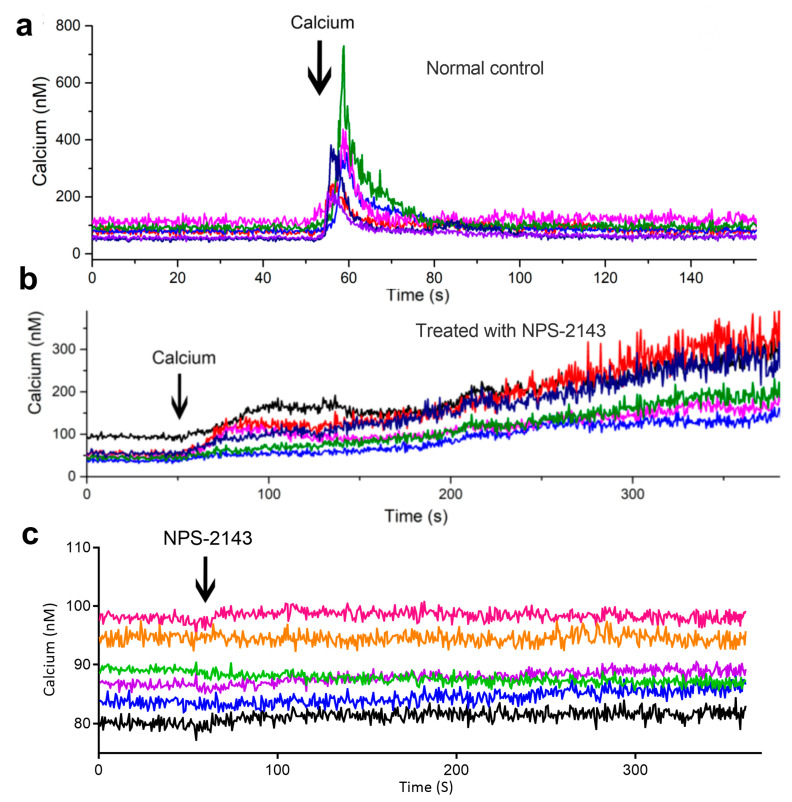
Intracellular calcium transient triggered by extracellular 7.2 mM Ca^2+^ in osteoblasts (**a**), treated with NPS-2143 (**b**), and (**c**) NPS-2143 control. Each colored line represents a cell sample.

**Figure 7 ijms-24-04991-f007:**
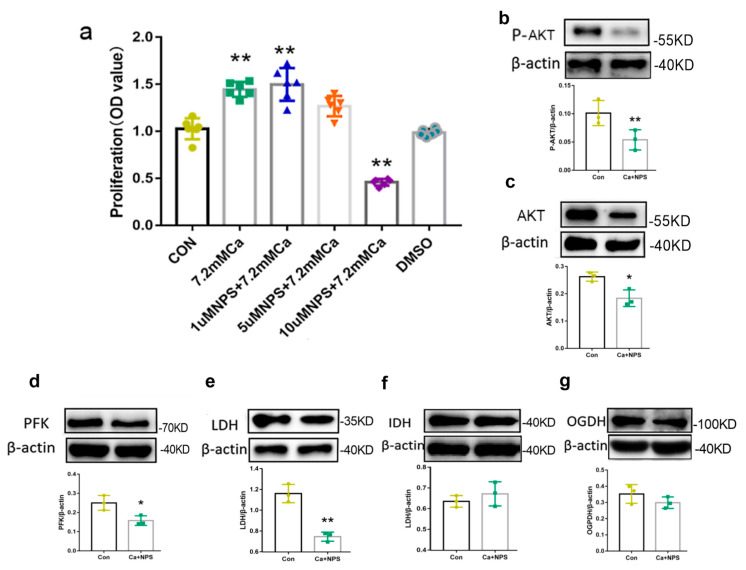
CaSR contributed to cell proliferation through AKT and then glycolic metabolism. (**a**) osteoblasts proliferation treated with extracellular 7.2 mM Ca^2+^ and NPS-2143. (**b**,**c**) changes of P-AKT and AKT induced by extracellular 7.2 mM Ca^2+^ and NPS-2143. (**d**–**g**) changes of OGDH, IDH, PFK and LDH induced by extracellular 7.2 mM Ca^2+^ and NPS-2143 * *p* < 0.05, ** *p* < 0.01 vs. Con.

## Data Availability

The data that support the findings of this study are available from the corresponding author upon reasonable request.

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
