# Peer review of "Extracellular Calcium-Induced Calcium Transient Regulating the Proliferation of Osteoblasts through Glycolysis Metabolism Pathways"

_ijms, 2023, doi:10.3390/ijms24054991_

Round 1

Reviewer 1 Report

The manuscript from Gao et al establishes the impact of extracellular calcium levels on osteoblast metabolism. Increasing extracellular calcium increased proliferation without impacting alkaline phosphatase activity, a surrogate for osteoblast differentiation. Increased extracellular calcium shifted the metabolic profile of osteoblasts, shifting toward aerobic glycolysis while decreasing ATP production and increasing both OCR and ECAR. Inhibition of the CaSR with NPS-2143 eliminated calcium-induced intracellular calcium peaks yet caused sustained increases in cytosolic calcium. Akt levels varied in response to treatment.

The manuscript is very brief and could benefit from increased description of the rationale, approaches, and outcomes. The absence of figure legends makes appreciation of the work especially challenging. Why do the authors conclude that CaSR-driven transients are required for proliferative and metabolic changes without considering the marked increased in intracellular calcium (Fig 6b) that occur when CaSR is inhibited? Further, control experiments with NPS2143 w/o changing calcium are missing. 

Author Response

Response to Reviewer #1's Comments

The manuscript from Gao et al establishes the impact of extracellular calcium levels on osteoblast metabolism. Increasing extracellular calcium increased proliferation without impacting alkaline phosphatase activity, a surrogate for osteoblast differentiation. Increased extracellular calcium shifted the metabolic profile of osteoblasts, shifting toward aerobic glycolysis while decreasing ATP production and increasing both OCR and ECAR. Inhibition of the CaSR with NPS-2143 eliminated calcium-induced intracellular calcium peaks yet caused sustained increases in cytosolic calcium. Akt levels varied in response to treatment.

The manuscript is very brief and could benefit from increased description of the rationale, approaches, and outcomes. The absence of figure legends makes appreciation of the work especially challenging. Why do the authors conclude that CaSR-driven transients are required for proliferative and metabolic changes without considering the marked increased in intracellular calcium (Fig 6b) that occur when CaSR is inhibited? Further, control experiments with NPS2143 w/o changing calcium are missing. 

Response: Thank you for your comments. The figure legends have been added into the revised manuscript.

As far as we know, the mechanism to stimulation of calcium signalling mainly due to stimulate phospholipase C (PLC), which hydrolyzes phosphatidylinositol 4, 5-diphosphate (PIP2) into second messenger diacylglycerol and inositol triphosphate (IP3). Among them, IP3 binds to the IP3 receptor of ER to promote the release of intracellular Ca2+ into the cytoplasm, while DAG activates a cascade of protein kinase C (PKC) and mitogen-activated protein kinase (MAPK) (Kifor et al., 2001). Therefore, [Ca2+]i one short spike calcium transient is triggered, but no secondary release of ER is caused after the unimodal oscillation is triggered, probably because AKT interferes with the hydrolysis of PIP2 and consumes part of it. PIP2 can also be phosphorylated to PIP3 with the activation of class I PI3K enzymes, which can bind to AKT in response to extracellular signals.

Extracellular high calcium lead to the increase of intracellular calcium ion, after intracellular ATP decreases instantly. Consumption of ATP could not increase calcium ion again, but reabsorbed into endoplasmic reticulum. Later, the acute glycolysis pathway is enhanced under the action of high calcium (it may also be enhanced through HIF (Denko, 2008; Semenza, 2004) to replenish ATP.

So there may be two reasons for “CaSR-driven transients are required but not the gradual increasing of intracellular calcium”.

1)The single-peak transient of calcium may be a signal mechanism that causes a cascade reaction of intracellular signals, and the cascade reaction of intracellular signals will not occur after the single-peak transient disappears. Many studies have reported that there are many "decoder" molecules in the changes of intracellular calcium transients, such as NF-kB, MAPK, and NFAT (Smedler & Uhlen, 2014; Tao et al., 2011). After the instantaneous increase of calcium ions, the rapid combination and dissociation of calcium ions through EF-hands or other structures may also be one of the reasons. The process of rapid combination and dissociation of calcium with enzymes or other molecules will not exist in the process of gradually increase. It should be meeting the glycolysis energy supply process as well as supply the increase of aerobic oxidation.

2)When extracellular calcium causes intracellular calcium to rise a single-peak transient through CaSR, it will cause (cell cycle increase) the transient response of glycolytic enzyme to increase in the cell, thus causing the increase of intracellular glycolysis, which can increase more energy for cell proliferation. The slow increase will also lead to the cell metabolism not inclined like the increase of glycolysis, and the calcium ion will also lead to the enhancement of aerobic oxidation. The instantaneous increase of calcium may first supply the needs of glycolysis enhancement, and then enter the absolute non-response period of intracellular calcium. Together, extracellular calcium can cause the transient of intracellular calcium through CaSR to promote proliferation.

We also added the sentence in line 18/page 15 “In this study, a [Ca2+]i transient was triggered by high extracellular Ca2+ concentrations (via CaSR) but not the gradual increase in [Ca2+]I levels. First, the single-peak transient of calcium may be a signaling mechanism which causes a cascade of intracellular signals. Moreover, this cascade reaction does not occur after the single-peak calcium transient disappears. Studies have reported the existence of numerous "decoder" molecules associated with changes in intracellular calcium transients, such as NF-kB, MAPK, and NFAT (Smedler & Uhlen, 2014; Tao et al., 2011). Second, when extracellular calcium concentration causes intracellular calcium levels to rise and generate a single-peak transient through CaSR, the transient response of glycolytic enzyme increases inside the cell. This promotes intracellular glycolysis to yield more energy for cell proliferation. The gradual increase in [Ca2+]i concentration promotes both glycolysis and aerobic oxidation, but not inclined to the increase of glycolysis. It has been reported that the increase in extracellular calcium levels stimulates DNA synthesis in osteoblasts and boosts their proliferation in vitro (Yamaguchi et al., 1998), which is consistent with the results of this study.”

Kifor, O., MacLeod, R. J., Diaz, R., Bai, M., Yamaguchi, T., Yao, T., . . . Brown, E. M. (2001). Regulation of MAP kinase by calcium-sensing receptor in bovine parathyroid and CaR-transfected HEK293 cells. Am J Physiol Renal Physiol, 280(2), F291-302. doi:10.1152/ajprenal.2001.280.2.F291

Denko, N. C. (2008). Hypoxia, HIF1 and glucose metabolism in the solid tumour. Nature Reviews Cancer, 8(9), 705-713. doi:10.1038/nrc2468

Semenza, G. L. (2004). Hydroxylation of HIF-1: Oxygen sensing at the molecular level. Physiology, 19, 176-182. doi:10.1152/physiol.00001.2004

Smedler, E., & Uhlen, P. (2014). Frequency decoding of calcium oscillations. Biochim Biophys Acta, 1840(3), 964-969. doi:10.1016/j.bbagen.2013.11.015

Tao, R., Sun, H. Y., Lau, C. P., Tse, H. F., Lee, H. C., & Li, G. R. (2011). Cyclic ADP ribose is a novel regulator of intracellular Ca2+oscillations in human bone marrow mesenchymal stem cells. Journal of Cellular and Molecular Medicine, 15(12), 2684-2696. doi:10.1111/j.1582-4934.2011.01263.x

Yamaguchi, T., Chattopadhyay, N., Kifor, O., Butters, R. R., Jr., Sugimoto, T., & Brown, E. M. (1998). Mouse osteoblastic cell line (MC3T3-E1) expresses extracellular calcium (Ca2+o)-sensing receptor and its agonists stimulate chemotaxis and proliferation of MC3T3-E1 cells. J Bone Miner Res, 13(10), 1530-1538. doi:10.1359/jbmr.1998.13.10.1530

We added the control experiments with NPS-2143 changing calcium in the figure 6c.

Fig. 6 Intracellular calcium transient triggered by extracellular 7.2 mM Ca2+in osteoblasts (a), treated with NPS-2143 (b), and (c) NPS-2143 control.

Reviewer 2 Report

The work by Xiaohang Gao et al., entitled "Extracellular calcium-induced calcium transient regulating the proliferation of osteoblasts through glycolysis metabolism pathways " presents an interesting study on the effects of high concentrations of extracellular calcium in bone microenvironment during remodelling process. Using an osteoblastic cell line (MC3T3-E1) as a model, this in vitro study demonstrates how high concentrations of calcium in the extracellular medium cause a transient increase in intracellular calcium, which in turn activates different metabolic pathways in the cell. Specifically, they focus on the activation of the glycolytic pathway through the AKT signalling pathway, and show how this fact promotes osteoblastic proliferation (MC3T3-E1 cells) and ultimately osteosynthesis during the bone remodelling process.

The study is well designed and correctly executed. The experimental approaches are valid and the results and conclusions obtained by the authors support the objectives of the work. However, there are several considerations that I would like the authors clarify and incorporate into the paper so that it is suitable for publication.

1) On the one hand, it must be taken into account that the authors work with an established cell model (MC3T3-E1 cell line), a well-established clonal osteogenic cell line, which provides an excellent model for the study of gene expression patterns in osteoblast differentiation. However, it is still a cellular model that might not be fully representative of osteoblast behaviour in vivo. Therefore, the authors should take this fact into account and include a section about it in the paper discussion.

2) In relation to the previous point, it would be interesting to verify that the behaviour and cellular response shown by MC3T3-E1 cells is also produced when primary cultures of osteoblasts are subjected to the same concentrations of extracellular Ca2+. It would be interesting to do the same experiments in osteoblast primary culture or in osteoblast differentiated from bone marrow MSC. Have the authors performed these experiments on primary cultures of osteoblasts or on osteoblasts differentiated from MSCs?

3) Why has the extracellular Ca2+ concentration of 7.2 mM been used? I imagine that this concentration is in the range that exists in the bone microenvironment during resorption processes, but I imagine that the concentration can vary within a range during this process. Why have different extracellular calcium concentrations not been tested? Could different responses be obtained with extracellular calcium concentrations below or above that used? It would be interesting to address this point in the paper discussion.

4) In the conclusion paragraph, Line 290, the authors cited “Our results demonstrated the role of high-level extracellular calcium in the microenvironment during bone remodeling” this sentence should be reformulated, since the authors show one of the effects of extracellular calcium in the microenvironment during bone remodelling , but it is quite probable that it also influences other aspects of cell behaviour related to bone remodelling, so it should not be stated that it is the only role that performs, as this conclusion affirms.

5) Finally, I don't know if it is an error but there are no figure captions in the manuscript that I have reviewed. They should be included since there are figures that are difficult to interpret, such as figure 2, that shows the metabolomics data.

Author Response

Response to Reviewer #2's Comments

The work by Xiaohang Gao et al., entitled "Extracellular calcium-induced calcium transient regulating the proliferation of osteoblasts through glycolysis metabolism pathways " presents an interesting study on the effects of high concentrations of extracellular calcium in bone microenvironment during remodelling process. Using an osteoblastic cell line (MC3T3-E1) as a model, this in vitro study demonstrates how high concentrations of calcium in the extracellular medium cause a transient increase in intracellular calcium, which in turn activates different metabolic pathways in the cell. Specifically, they focus on the activation of the glycolytic pathway through the AKT signalling pathway, and show how this fact promotes osteoblastic proliferation (MC3T3-E1 cells) and ultimately osteosynthesis during the bone remodelling process.

The study is well designed and correctly executed. The experimental approaches are valid and the results and conclusions obtained by the authors support the objectives of the work. However, there are several considerations that I would like the authors clarify and incorporate into the paper so that it is suitable for publication.

1) On the one hand, it must be taken into account that the authors work with an established cell model (MC3T3-E1 cell line), a well-established clonal osteogenic cell line, which provides an excellent model for the study of gene expression patterns in osteoblast differentiation. However, it is still a cellular model that might not be fully representative of osteoblast behaviour in vivo. Therefore, the authors should take this fact into account and include a section about it in the paper discussion.

Response: Thank you for your good comments. We have added sentence in line 11/page 19 “Second, our results were generated from experiments on MC3T3-E1 cell lines in vitro. Although MC3T3-E1 is a widely used osteoblast cell line (Izumiya et al., 2021), which can mimic the physiological activity of typical osteoblasts (Quarles, Yohay, Lever, Caton, & Wenstrup, 1992), this study did not conduct animal experiments to verify the effects of high extracellular calcium levels on osteoblasts and bone regeneration in vivo.”

Izumiya, M., Haniu, M., Ueda, K., Ishida, H., Ma, C., Ideta, H., . . . Haniu, H. (2021). Evaluation of MC3T3-E1 Cell Osteogenesis in Different Cell Culture Media. Int J Mol Sci, 22(14). doi:10.3390/ijms22147752

Quarles, L. D., Yohay, D. A., Lever, L. W., Caton, R., & Wenstrup, R. J. (1992). Distinct proliferative and differentiated stages of murine MC3T3-E1 cells in culture: an in vitro model of osteoblast development. Journal of Bone and Mineral Research, 7(6), 683-692. doi:10.1002/jbmr.5650070613

2) In relation to the previous point, it would be interesting to verify that the behaviour and cellular response shown by MC3T3-E1 cells is also produced when primary cultures of osteoblasts are subjected to the same concentrations of extracellular Ca2+. It would be interesting to do the same experiments in osteoblast primary culture or in osteoblast differentiated from bone marrow MSC. Have the authors performed these experiments on primary cultures of osteoblasts or on osteoblasts differentiated from MSCs?

Response: We have not carried out similar experiments in other cell lines, only in the osteoblast cell line, but we think that MC3T3-E1 cell lines was the widely used and typical osteoblast cell lines. MC3T3-E1 cells can well simulate the proliferation and differentiation of osteoblasts in vivo. In the following work, we will explore the specific role of calcium transient by extracellular Ca2+ through CaSR in animal experiments. Studies indicated that MC3T3-E1 cells are one of the most commonly used osteoblast-like cell lines for evaluation of osteogenesis (Hwang & Horton, 2019; Izumiya et al., 2021). Early study reported that MC3T3-E1 cells display a time-dependent and sequential expression of osteoblast characteristics analogous to in vivo bone formation (Quarles et al., 1992). There are other studies using the MC3T3-E1 cells as typical osteoblasts to research the osteoblast biology in the experiment (Gu, Fu, Yuan, & Liu, 2017; Izumiya et al., 2021; Kim et al., 2005; Wei et al., 2020). Thus, MC3T3-E1 cells were used in this study.

Gu, C., Fu, L., Yuan, X., & Liu, Z. (2017). Promoting Effect of Pinostrobin on the Proliferation, Differentiation, and Mineralization of Murine Pre-osteoblastic MC3T3-E1 Cells. Molecules, 22(10). doi:10.3390/molecules22101735

Hwang, P. W., & Horton, J. A. (2019). Variable osteogenic performance of MC3T3-E1 subclones impacts their utility as models of osteoblast biology. Sci Rep, 9(1), 8299. doi:10.1038/s41598-019-44575-8

Izumiya, M., Haniu, M., Ueda, K., Ishida, H., Ma, C., Ideta, H., . . . Haniu, H. (2021). Evaluation of MC3T3-E1 Cell Osteogenesis in Different Cell Culture Media. Int J Mol Sci, 22(14). doi:10.3390/ijms22147752

Kim, S. W., Her, S. J., Park, S. J., Kim, D., Park, K. S., Lee, H. K., . . . Kim, S. Y. (2005). Ghrelin stimulates proliferation and differentiation and inhibits apoptosis in osteoblastic MC3T3-E1 cells. Bone, 37(3), 359-369. doi:10.1016/j.bone.2005.04.020

Quarles, L. D., Yohay, D. A., Lever, L. W., Caton, R., & Wenstrup, R. J. (1992). Distinct proliferative and differentiated stages of murine MC3T3-E1 cells in culture: an in vitro model of osteoblast development. Journal of Bone and Mineral Research, 7(6), 683-692. doi:10.1002/jbmr.5650070613

Wei, W., Liu, S., Song, J., Feng, T., Yang, R., Cheng, Y., . . . Hao, L. (2020). MGF-19E peptide promoted proliferation, differentiation and mineralization of MC3T3-E1 cell and promoted bone defect healing. Gene, 749, 144703. doi:10.1016/j.gene.2020.144703

3) Why has the extracellular Ca2+ concentration of 7.2 mM been used? I imagine that this concentration is in the range that exists in the bone microenvironment during resorption processes, but I imagine that the concentration can vary within a range during this process. Why have different extracellular calcium concentrations not been tested? Could different responses be obtained with extracellular calcium concentrations below or above that used? It would be interesting to address this point in the paper discussion.

Response: This part of the experiment has been completed, but it was not shown in the manuscript. The concentration-response curve of the Ca2+ treatments for proliferation were added in the revised manuscript as supplementary figure.

We added sentence in line 10/page 9 : “MC3T3-E1 cells were treated with various concentrations (3.6-9 mM) of extracellular calcium for 72 h and their proliferation and differentiation were determined. The 7.2 mM extracellular calcium concentration promoted the proliferation of osteoblasts more effectively than the other concentrations assayed (Supplementary Fig. S1).”

Fig. S1 (a) Proliferation of osteoblast cells induced by different concentration-response of the extracellular Ca2+ P. *P < 0.05, vs. Con group. (b) Differentiation of osteoblast cells induced by different concentration-response of the extracellular Ca2+. *P < 0.05, **P < 0.01, vs. Con group

4) In the conclusion paragraph, Line 290, the authors cited “Our results demonstrated the role of high-level extracellular calcium in the microenvironment during bone remodeling” this sentence should be reformulated, since the authors show one of the effects of extracellular calcium in the microenvironment during bone remodelling , but it is quite probable that it also influences other aspects of cell behaviour related to bone remodelling, so it should not be stated that it is the only role that performs, as this conclusion affirms.

Response: According reviewer’s comments, we have revised the paragraph as (Line 1/page 20) “In conclusion, our current work extended the findings of previous studies and verified that CaSR was involved in the induction of [Ca2+]i transients by high extracellular calcium levels, which in turn promoted glycolysis metabolisms via AKT activation, and ultimately increased osteoblast proliferation.”

5) Finally, I don't know if it is an error but there are no figure captions in the manuscript that I have reviewed. They should be included since there are figures that are difficult to interpret, such as figure 2, that shows the metabolomics data.

Response: We are sorry for missing the figure captions. We have added the figure captions in the figures-word.

Round 2

Reviewer 2 Report

The authors have adequately incorporated the suggestions and have taken into account the comments, which in my opinion has improved the paper, being now suitable for publication.